# Coffee Silver Skin—Health Safety, Nutritional Value, and Microwave Extraction of Proteins

**DOI:** 10.3390/foods12030518

**Published:** 2023-01-23

**Authors:** Vedran Biondić Fučkar, Angela Božić, Anita Jukić, Adela Krivohlavek, Gordana Jurak, Ana Tot, Sonja Serdar, Irena Žuntar, Anet Režek Jambrak

**Affiliations:** 1Faculty of Pharmacy and Biochemistry, University of Zagreb, 10000 Zagreb, Croatia; 2Faculty of Food Technology and Biotechnology, University of Zagreb, 10000 Zagreb, Croatia; 3Andrija Štampar Teaching Institute of Public Health, 10000 Zagreb, Croatia

**Keywords:** coffee silver skin (CS), health safety, nutraceutical, pesticides, heavy metals, fibre, proteins, microwave extraction, food industry

## Abstract

The aim of this research was to evaluate the health safety (concentrations of pesticide residues and heavy metals) and nutritional parameters (macro- and microminerals and crude fibre) of coffee silver skin (CS), as well to isolate proteins from this by-product using an optimised microwave extraction method. The CS by-product samples showed the highest amount of potassium, followed by calcium, magnesium, and sodium. Iron was found in the highest quantity among the microminerals, followed by copper, manganese, zinc, and chromium. The CS sample showed a large amount of fibre and a moderate quantity of proteins obtained by the optimised microwave extraction method. Four heavy metals (nickel, lead, arsenic, and cadmium) were detected, and all were under the permitted levels. Among the 265 analysed pesticides, only three showed small quantity. The results for the proteins extracted by microwave showed that the total protein concentration values ranged from 0.52 ± 0.01 mg/L to 0.77 ± 0.07 mg/L. The highest value of the concentration of total proteins (0.77 ± 0.07 mg/L) was found in the sample treated for 9 min, using a power of 200 W. Based on these results, it can be concluded that CS is a healthy and nutritionally rich nutraceutical that could be used in the production of new products in the food industry and other industries.

## 1. Introduction

Coffee silver skin (CS) is a thin tegument that directly covers the coffee seed. It consists of various components that can be used and treated in reprocessing. It is known that CS is a high-volume by-product of coffee roasting [1]. As a healthy raw material, it could be used in the production of new food products, for the purpose of enriching existing or new food products, or as a dietary supplement, functional food, and/or nutraceutical.

The major component in CS is dietary fibre (up to 68.5%), which includes two types of fibres: insoluble (46–56%) and soluble (8.2–11%) fibres [2,3,4,5,6,7,8]. Dietary fibre is one of the main nutritional factors contributing to human well-being. The European Food Safety Authority (EFSA) defines dietary fibres as non-digestible carbohydrates, including non-starch polysaccharides, resistant starch, oligosaccharides, and lignin [9]. Diets in the Western world are impoverished of fibres that have important beneficial physiological effects and are associated with the risk of non-communicable chronic diseases [10]. In the large intestine, dietary fibre is fermented by the microbiota, leading to the generation of short-chain fatty acids (SCFAs) that also contribute to human well-being. The high dietary fibre content of CS might benefit the intestine and gut microbiota, as well as overall health [11].

The second major component present in CS is protein (7.1–22%), followed by carbohydrates (9.5–14.5%) and fat (1.6–3%) [2,3,4,5,6,12,13,14,15]. CS is also a source of polyphenols, particularly chlorogenic acid (CGA) of which 5-O-, 3-O-, and 4-O-caffeoylquinic acids are the most relevant. This by-product also contains caffeine (1%) and melanoidins (5%) which are formed during the roasting process. The CS mineral composition consists mainly of potassium, magnesium, and calcium (about 5 g, 2 g, and 0.5 g per 100 g of silver skin). In addition to potassium, magnesium, and calcium, various other minerals, such as iron, manganese, copper, phosphorus, and aluminium, can be found in CS [5,7,8,16,17]. Recently, Lorbeer et al. calculated that 26 g of estimated daily intake of CS contained 32% of daily dose for potassium, 14–40% for magnesium, 13–26% for calcium, 50–200% for iron, and around 5% for zinc [8]. In addition, 1/12 of daily reference dose of vitamin E and less than 1% of daily dose of vitamins B2 and B3 were found in 26 g of estimated daily dose of SC [8].

Pesticide and heavy metal residue monitoring should be performed when CS samples are examined from a food safety standpoint to effectively protect consumer health. Some metals are harmful even at low concentrations, and they also accumulate in tissues over time [18]. Since the Earth’s origin, heavy metals have been found naturally on the Earth’s crust. The prime cause of heavy metal pollution in both terrestrial and aquatic environments is anthropogenic activity (metal mining, smelting, foundries, and other metal-based industries, as well as leaching of metals from different sources, such as landfills, waste dumps, excretion, livestock and chicken manure, runoffs, automobiles and roadworks), and the second cause is their use in the agricultural field (pesticides, insecticides, fertilisers, etc.) [19,20,21,22]. Additionally, studies showed that various pesticide residues have been found in coffee beans that expose consumers to dangerous agrochemicals [22,23,24,25]. Among pesticides and heavy metals are other contaminants such are 5-hydroxymethylfurfural, which is produced by the thermal decomposition of carbohydrates and suspected to be a genotoxic and carcinogenic agent. Polycyclic aromatic hydrocarbons, acrylamide, and phytosterol oxidation products were also detected but in low amount or only in traces [8].

One of the effective and environmentally friendly methods of extracting components from CS is microwave extraction (MW). The advantages of MW are shorter extraction time, faster heating, lower temperature gradient, and reduced use of solvents. The reduced use of solvents and the possibility of using water as a solvent make this method more environmentally friendly than the classic methods of extracting components from plant materials. Microwave-assisted extraction (MW) is a non-invasive, relatively new method of extracting bioactive components, mainly from plant materials [26]. Microwave extraction is highly dependent on the dielectric properties of the solvent and the sample, the solubility of the sample in the solvent, and the temperature of the process. Water and polar solvents that have a higher dielectric constant are mainly used for microwave extraction [27]. In addition to the dielectric constant, an important factor when choosing a solvent is the scattering factor, which shows the efficiency of the conversion of electromagnetic energy into heat [28]. Water has a higher dielectric constant than polar solvents, such as ethanol and methanol, but has a lower scattering factor. For this reason, a mixture of an organic solvent and water can be used for microwave extraction to obtain a solvent with a higher dielectric constant and a more significant scattering factor. In addition to the solvent, an important parameter for microwave extraction is temperature. A higher extraction temperature generally improves the extraction performance; however, too high a temperature can adversely affect the thermolabile components in the sample. Microwave extraction can be performed in a closed or open system. In closed systems, the extraction takes place in a closed vessel with uniform microwave heating, where it is preferable that the system is designed so that there is the possibility of pressure and temperature control. This type of system can achieve higher temperatures and, thus, more efficient extraction thanks to the creation of increased pressure in the vessel, which increases the evaporation point of the solvent [29]. However, in such a system, the possibility of treating many samples simultaneously is mostly limited. Open MW systems may be more suitable for thermolabile compounds, such as bioactive components from plant materials.

The purpose of this article is to evaluate the health safety and nutritional value of CS, the most abundant by-product of roasting coffee. The aim is to analyse the concentration of valuable macro- and microminerals and crude fibres, to isolate proteins from the CS samples using an optimised MW method, and to provide a protein analysis as well as determine the concentration of pesticide residues and heavy metals.

## 2. Materials and Methods

### 2.1. Samples

A sample of the by-products after coffee roasting, i.e., CS sample, was used in his research. The sample was obtained from a Croatian company involved in the production of coffee, tea, and snack products from Zagreb, Franck Ltd. The studied sample contains a mixture of *Coffea arabica* (Arabica) and *Coffea canephora* (Robusta) coffee, since the plant collects the by-products from all roasters, which are then converted into pellets. The sample of by-products was obtained in the form of pellets which were ground into powder for research purposes to facilitate the analysis. Before analysis, the pellets were grinded and homogenised. Each analysis was performed in six parallel probes.

### 2.2. Chemicals

The following chemicals were used in this research: aceton (Lach-Ner, Neratovice, Czech Republic); acetonitrile (Applichem, Darmstadt, Germany); ammonium formate (GramMol, Zagreb, Croatia); argon, 99.9995% (Messer, Bad Soden am Taunus, Germany), certified element reference material (CPAChem, Bogomilovo, Bulgaria); distilled water and nitric acid (Gram-Mol, Zagreb, Croatia); helium, 6.0 (Messer, Bad Soden am Taunus, Germany); methanol (GramMol, Zagreb, Croatia); mixture of magnesium sulphate, sodium chloride, and citrate salts (Citrate-Kit-01, BEKOlut, Bruchmühlbach-Miesau, Germany); mixture of magnesium sulphate salt, primary secondary amine, and GCB (PSA-Kit-06, BEKOlut, Bruchmühlbach-Miesau, Germany); sodium hydroxide (T.T.T., Sveta Nedjelja, Croatia); petrol ether (Applichem, Chicago, IL, USA); sulfuric acid (Chemistry, Zagreb, Croatia); and hydrogen peroxide (Gram-Mol, Zagreb, Croatia).

### 2.3. Methods for Determination of Target Compounds in CS Sample

#### 2.3.1. Determination of Pesticide Residues

The pesticide residue samples were prepared by using the Quick Easy Cheap Effective Rugged Safe method (QuEChERS) according to Anastassiades et al. [30]. After that, they were quantified by gas or liquid chromatography coupled to a triple quadrupole (QqQ) mass spectrometry (GC-MS/MS, LC-MS/MS). In brief, the prepared samples were injected and analysed on both instruments to determine as many pesticides as possible. The samples analysed by the gas chromatography (GCMS-TQ8050 NX, Nexis GC-2030, Shimadzu, Kyoto, Japan) were recorded in GC-MS Solution and the data were processed in LabSolutions Insight GCMS. The samples analysed by the liquid chromatography (UPLC-MS/MS, Xevo TQ MS, Waters, Milford, CT, USA) were recorded in MassLynx, and the data were processed in TargetLynxXS. The chromatographic conditions under which the gas and liquid chromatography measurements were performed are shown in Table 1. The obtained chromatograms were analysed based on the retention time (RT) of the components in the sample and the standard, and the individual pesticides in the samples were qualitatively identified by using two-ion transitions for LC-MS/MS determination and three-ion transitions for the MRM of each pesticide for GC-MS/MS determination. The ratio of the area below the peak in the sample and the individual standard provide the quantitative information, i.e., the concentration of pesticides which are present in the sample [31].

#### 2.3.2. Determination of Valuable Minerals and Heavy Metals

The CS sample was first degraded and prepared for measurement and then measured on an inductively coupled plasma instrument with a mass spectrometer as a detector (ICP-MS) [32]. Briefly, 0.5 g of the crushed and homogenised CS sample was first weighed into the Teflon cuvette of the microwave device. Then, 1 mL of water was added to the sample, weighed, and left for 1 h. After that, 3 mL of concentrated nitric acid and 1 mL of 30% hydrogen peroxide were added. The cuvette with the sample was closed and placed in a microwave decomposition drum (ultraWAVE, Milestone, Sorisole, Italy). Upon the completion of the decomposition procedure, the sample was cooled, the cuvette was opened, and the clear solution was poured through a glass funnel and quantitatively transferred to a 25 mL volumetric flask while being rinsed with deionized water. The sample was prepared and ready for measurement when it was without suspended particles. Firstly, a calibration curve with 3 standard solutions was made, after which the measurement of the samples was conducted. All the samples were prepared and analysed in two parallels. An inductively coupled plasma with a mass detector (ICP-MS 7900, Agilent, Santa Clara, CA, USA) and an autosampler (according to the conditions in Table 2) was used. Inert gases were used for the operation of the instrument: argon with a purity of 99.9995% and helium with a purity of 6.0. Individual standards were used to prepare the calibration curves. A linearity of ≥0.999 was achieved for the calibration curve of each element. An internal standard (Bi, Sc, Y, Ge) with a concentration of 100 µg/L was used.

#### 2.3.3. Determination of Nutritional Value of CS Dietary Fibre (CSDF)

The determination of crude fibre in this study was determined using the method for the determination of the mass fraction of crude fibre in food from AOAC 962.09 [33]. Crude fibre is defined as the residue after the process of decomposition, drying, and incineration of a sample.

The CS samples (6 parallel probes) were tested in six parallels by first weighing 0.95–1.00 g of each sample into labelled filter bags. This should be performed very carefully so as not to catch the sample above the part of the bag that would be welded later by means of a welding device at about 4 mm from its top. It was also necessary to weigh one empty bag that would serve as a blank. All the samples were placed in a beaker and poured over with petroleum ether to extract the fat. The overflowing bags were left to stand for 10 min and then they were dried in a desiccator. The bags were shaking to evenly distribute the sample over the bag and to break up lumps. The bags were fitted in a frame and placed in a raw fibre analytical device (Ankom 2000, Ankom, Macedon, NY, USA). After decomposition in the raw fibre analytical device, the bags were cool, and excess water was carefully squeezed out of the bags. The bags were immersed in acetone and left to stand for 3–5 min, after which they were dried. When they were completely dry, they were placed in the oven at 102 ± 2 °C for 2–4 h. After drying, they were cooled and weighed. The ash was then determined by incineration in weighed porcelain pots at 600 ± 15 °C for 2 h. After cooling, it was weighed again. The calculation of raw fibres is performed according to the following equations:$$C_{1}=\frac{m\;\left(bag\;after\;drying\right)-m\;\left(ash\right)}{m\;\left(bag\right)}\;m\;\left(ash\right)=m\;\left(pot\;after\;annealing\right)-m\;\left(pot\right)\;w\;\left(fibers\right)=\frac{\left\{\left[m\;\left(nafter\;drying\right)-m\;\left(ash\right)\right]-\left[m\;\left(bag\right)\times C_{1}\right]\right\}}{m\;\left(sample\right)}\times 100$$

#### 2.3.4. Microwave Extraction of Total Proteins—Impact on Total Protein’s Proportion

Microwave extraction (MW) is a non-invasive, relatively new method for extracting bioactive components, mainly from plant materials as described previously [27,28,29]. Microwave extraction of the CS samples was carried out using deionized water as a solvent. The extraction was performed on a microwave laboratory synthesis device StartSYNTH (Milestone, Italy). Before the extraction, the design of the experiment was created in the program STATGRAPHICS Centurion (StatPoint technologies, Inc., Warrenton, VA 20186, USA). After the extraction, the obtained results were processed in the same program. Using STATGRAPHICS, an analysis of variance (ANOVA) was performed, which determined the statistical significance of power and time parameters on the values of total proteins and polyphenols. Those parameters which *p*-value < 0.05 were taken as statistically significant parameters. Diagrams of the response surfaces of the experimental data and the equations of the experimental models were also created. Extraction optimisation was also carried out, in which the optimal extraction parameters were determined.

The determination of protein concentration by the Lowry method [34] was carried out spectrophotometrically. From a standard protein solution with a concentration of 200 mg/mL, solutions of known protein concentrations of 0.02, 0.04, 0.06, 0.08, 0.1, and 0.12 mg/L were prepared in a 10 mL volumetric flask. After measuring the absorbance, a calibration diagram was created.

## 3. Results

### 3.1. Pesticide Residues

In this work, we attempted to analyse pesticide residues in CS by analysing as many active substances as possible (635 active substances on LC-MS/MS or GC-MS/MS). A total of 265 pesticides passed the validation experiments in accordance with the guidance SANTE/12682/2019, and their linearity, repeatability, reproducibility, and utilization of the extraction procedure were tested. In total, 265 pesticides were successfully analysed in the CS samples. Pesticide residues were determined by LC-MS/MS and/or GC-MS/MS. The main reason is that these technologies are not equally sensitive to all pesticides. Our goal was to test as many pesticides as we could, but due to the heavy matrix in the CS samples and the suppressions of ions, 128 pesticides were successfully quantified by GC-MS/MS and 137 pesticides by LC-MS/MS. Table 3 shows three pesticides which have been detected and quantified in a CS sample. The detected and quantified pesticides are flutriafol, imidacloprid, and piperonyl butoxide. The results are presented as the average of six parallel analyses.

### 3.2. Valuable Minerals and Heavy Metals

The ICP-MS determined various metals that were classified into the categories of macrominerals, microminerals and heavy metals. Macrominerals are considered substances that the human body needs in large quantities, unlike microminerals that are present in the body in small quantities; macrominerals are essential for a vast array of physiological functions, as well as for normal growth and development. In our CS sample, potassium was found in the highest amount (K; 24,311 ± 764 mg/kg), followed by calcium (Ca; 10,569 ± 210 mg/kg), magnesium (Mg; 4672.5 ± 88.5), and sodium (Na; 329 ± 79 mg/kg). Among the microminerals, the CS sample contained iron (Fe) in the highest quantity (567.50 ± 12.50 mg/kg), followed by copper (Cu; 74.90 ± 0.60 mg/k), manganese (Mn; 37.55 ± 0.45 mg/kg), zinc (Zn; 16.80 ± 0.70 mg/kg), and chromium (Cr; 1.53 ± 0.04 mg/kg). Heavy metals are a group of metals with a density greater than 5 g/cm^3^ [19], so this group includes a list of metals, including some microminerals which are categorized into a separate group in this paper. Due to their persistence, high toxicity, and tendency to accumulate in the ecosystem, heavy metals are dangerous for living organisms [35]. Four heavy metals, including nickel, lead, arsenic, and cadmium, were detected in a CS sample and are shown in Table 4. The results are presented as the average of six parallel analyses.

### 3.3. Dietary Crude Fibres

Crude fibre is only a part of the dietary fibres, more precisely the part of insoluble fibres that remain after laboratory treatment with acid and alkali. This group mainly includes cellulose, hemicellulose, and lignin. Food contains more dietary fibre than crude fibre, but there is no quantitative relationship between these two types of fibres [36]. In the studied CS sample, 31.97 ± 0.61 g/100 g of crude fibres was detected. Given that it is only part of the insoluble fibre, the total fibre is much more. The representation of fibres in the sample is uniform. Considering the large amount of fibre, the skin of the coffee bean has great potential to be used as a raw material in the development of functional food. The results are presented as the average of six parallel analyses.

### 3.4. Microwave Extraction (MW) Conditions—Impact on the Proportion of Total Proteins in the Samples

Total protein concentration values range from 0.52 ± 0.01 mg/L to 0.77 ± 0.07 mg/L (Table 5). The highest value of the concentration of total proteins (0.77 ± 0.07 mg/L which is 0.77 mg/L i.e., 0.007%.) was found in the sample M9, which was treated for 9 min with a microwave power of 200 W. Such a result is expected since it is the sample that was treated for the longest time (9 min) with the highest applied microwave power (200 W). The second-best result of protein concentration (0.69 ± 0.01 mg/L) was obtained for the sample M6, which was treated with a power of 150 W for 9 min. A similar value of 0.66 ± 0.06 mg/L was obtained for the sample M8 treated for 6 min, with a power of 200 W. Other samples showed lower values of total protein concentrations, ranging from 0.52 ± 0.01 mg/L (M2) to 0.59 ± 0.01 mg/L (M7). The three samples treated with 100 W of microwave power showed almost equal total protein concentrations of 0.52 ± 0.01 mg/L (M2), 0.52 ± 0.08 mg/L (M3), and 0.53 ± 0.00 mg/L (M1), respectively. This means that when using low microwave power, prolonging the extraction time does not significantly increase the proportion of extracted proteins. The results are presented as the average of six parallel analyses.

To test the statistical significance of each factor of MW, an analysis of variance (ANOVA) was performed by comparing the mean square value with the experimental error estimate. The results are shown in Table 6. Three factors, including microwave power (A), extraction time (B), and their interactions (AB), have a *p*-value < 0.05, which means that these three factors have a significant effect on the change in the proportion of total proteins. This is important because it shows that when optimising the process and planning future experiments, we must consider the microwave power and the extraction time because both factors significantly affect the concentration of total proteins in the extract. The R-squared value is 98.901%, which means that 98.901% of the variability of the total protein concentration results can be explained by the experiment being set up in this way.

Additionally, the effects of power and time on the proportion of total proteins during the microwave extraction procedure is tested statistically and presented by a counter plot (Figure 1).

The diagram in Figure 1 gives a graphic overview of the statistical processing. It is evident that both microwave power and extraction time have an influence on the concentration of total proteins. In the contour diagram, the highest concentration values of total proteins are in the upper right corner. Such displays of response surfaces give us the possibility of easier process planning and prediction of future measurements, because relatively small changes in the concentration of total proteins with changes in microwave power and extraction time are shown.

## 4. Discussion

Our study provides data on the chemical characteristics of CS, especially those that pose an impact on human health. It is known that residual pesticides and heavy metals (Table 3 and Table 4) may affect human health; thus, their concentrations must be checked and measured to validate the compliance of products for human consumption.

The results of the analysis of pesticide residues and heavy metals in our study showed that the CS samples are safe for human consumption and, thus, can be used as ingredient for other products, such as functional foods, food supplements and cosmetics. All pesticides are below the maximum permitted level of pesticide residues in food and the studied CS does not pose a risk to human health. The detected pesticide flutriafol (Table 3) is a contact and systemic fungicide belonging to the triazole class. It is used on various cereals and for seed treatment. Its mechanism of action is inhibition of ergosterol biosynthesis and, thus, disruption of fungal cell wall synthesis [37]. Imidacloprid (Table 3) belongs to a group of neonicotinoid/nitroguanidine compounds used as insecticides. It acts as an antagonist by binding to postsynaptic nicotinic receptors in the central nervous system of insects. It is widely used on almost all crops [38]. Piperonyl butoxide (Table 3) has been used in insecticidal formulations for more than 50 years and always in combination with other insecticides, such as pyrethrins, synthetic pyrethroids, organophosphates, and carbamates. Piperonyl butoxide itself is not an insecticide, but it is a very good synergist and enhances its action when in combination with other insecticides. It allows more toxins to reach the target molecule, which increases the mortality of the target organism [39]. Finally, when comparing the results of research on coffee beans and research on the CS [40] and the results obtained in this study, it shows that the by-product of coffee contains much less pesticides and in very small quantities. The reason for this is that the coffee bean undergoes a process of roasting at high temperatures before commercialisation, which leads to the decomposition of pesticides [41]. Additionally, Nolasco et al. [17] in their research on three classes of SC samples (two main varieties of SC obtained by roasting *Coffea arabica* and *Coffea robusta* green beans and their blend) found that all values detected were at <LOQ. They related these results to the botanical differences between the two coffee variates, way of production of coffee, and difference in the root depth of the variates.

In our studied CS sample, nickel has the highest concentration, while lead, arsenic, and cadmium are detected in much lower concentrations (Table 4). According to Commission Regulation no. 1881/2006 [42], the CS sample could be classified as a dietary supplement, and it could be used for this purpose in the future. The same Regulation states that the maximum permitted level of lead in food additives is 3.0 mg/kg. Since 0.249 ± 0.077 mg/kg of lead is detected in the CS sample, it can be established that the detected amount of lead is not dangerous for the human body and does not pose a danger to human health. Furthermore, the Regulation defines that the maximum permitted amount of cadmium in food supplements may be 1.0 mg/kg. In this paper, it was proved that cadmium in the sample of CS has an amount of 0.106 ± 0.002 mg/kg, which is within the permissible values, and cadmium does not pose a danger to human health. Arsenic is a naturally occurring element in the Earth’s crust. It is also present in many foods due to absorption from soil and water. Rice, compared to other foods, can absorb higher amounts of arsenic, and since its consumption is high worldwide, it can contribute to high arsenic exposure. To protect consumers from overexposure, the Codex Alimentarius Commission has recommended that arsenic levels in rice should not exceed 0.2 mg/kg [43]. The main source of arsenic in diet comes from water that has been contaminated with arsenic. Therefore, the World Health Organization (WHO) has set a guideline concentration of 10 µg/L in drinking water [44]. Because of the difficulties with the analysis of (inorganic) arsenic in several food commodities, the maximum allowed levels for arsenic had initially been set for only rice and derived products. Once reliable analytical methods became available, a monitoring campaign for arsenic covering the period 2016–2018 was organised. The aim of this monitoring campaign is to generate reliable occurrence data that can be used to correctly evaluate the need for setting additional maximum levels for other food commodities [45]. Regarding arsenic quantity, Hackethal et al. [46] performed the first German total diet study and found the highest mean levels in the main food group “fish, fish products and seafoods”, followed by “vegetable and vegetable products”, “legumes, nuts, oil seeds and spices”, and “food products for young population”. This study did not include CS in their research. However, the main food group in their study which was very close and comparable to our type of sample, i.e., “coffee, cocoa, tea and infusions”, showed a lower maximum level of total arsenic, while other groups, e.g., “grains and grains-based products”, showed a higher level. Nickel is another metal for which there is currently no maximum residue level (MRL), so it is not possible to know exactly what the maximum allowable level of nickel in food is. In 2015, EFSA summarised the results of various studies to determine the concentrations of nickel in food [47]. According to this report, the highest concentrations of nickel were found in mushrooms, cocoa, cocoa products (>10 mg/kg), beans (9.8 mg/kg), soybeans (5.2 mg/kg), soy products (5.1 mg/kg), nuts (3.6 mg/kg), peanuts (2.8 mg/kg), cereals (2.3 mg/kg), buckwheat (2.0 mg/kg), and oats (1.8 mg/kg). In addition, nickel was found in beer (30 µg/L) and in wine (100 µg/L). EFSA also added that the factors influencing the concentration of nickel in food are growth conditions (much higher concentrations of nickel were found in contaminated soil areas) and the way food is prepared (impact of cooking utensils) [48]. In the CS sample, nickel levels of 2.495 ± 0.055 mg/kg were determined in our research. Since there is no maximum acceptable amount of nickel in food, it cannot be determined whether this amount is too high. However, according to the EFSA report, it can be noticed that nickel is normally present in food in high concentrations. Currently, there are no maximum levels in the EU legislation for nickel as a contaminant in foodstuffs. The maximum limit for nickel in natural mineral water is regulated in the EU by Commission Directive 2003/40/EC [49]. In this Directive, nickel is listed in Annex I among the constituents naturally present in natural mineral water, with a maximum limit of 20 μg/L. Nickel can be released from coffee machines, generally at a low concentration; however, concentrations above the specific release limit (SRL) (up to 780 μg/kg) have been reported after decalcification in two out of eight machines [50]. The authors pointed to the importance of sufficient rinsing after decalcification. Only low releases (maximum: 4.9 μg/L) were detected in the same study when water was boiled in electric kettles. When comparing our results of heavy metals with other studies [7,17] containing mixed CS samples (*C. arabica* and *C. canephora*), we found more lead and cadmium, slightly more arsenic, and almost the same amount of nickel. Additionally, in comparison to another study [51], we found the same amounts of arsenic and cadmium, and very close quantities of lead and nickel. A study by Ballesteros et al. [5] found lower quantities of nickel and lead while cadmium was slightly higher. The highest amount of nickel (3.17 ± 0.23 mg/kg) was found in a study with a mixed CS sample by Mrtuscelli et al. [52]. All these differences were probably related to the composition of the mixed SC samples (% of both varieties), the way of herb cultivation (ground, water, used fertilisers, and pesticides), and conditions of coffee production (used coffee blends and roasting temperatures).

The determined concentrations of macrominerals (potassium, calcium, sodium, and magnesium) showed that the concentrations of magnesium (4672.5 ± 88.5 mg/kg), calcium (10,569 ± 210 mg/kg), and potassium (24,311 ± 764 mg/kg) partially meet daily needs according to FDA recommendations [53], while sodium concentration (329 ± 79 mg/kg) is significantly lower. Therefore, if CS were used to enrich food products, in the production of new products, or as a food supplement, depending on the amount taken, it would satisfy part of the body’s daily need for macrominerals, especially magnesium and calcium, followed by potassium and slightly less sodium. Our macromineral results are in concordance with the data of Costa et al. [16], where the CS sample in their study contains the highest amounts of potassium (5 g/100 g), magnesium (2 g/100 g) and calcium (0.5 g/100 g). Ballesteros et al. [5] reported that their CS sample contains the highest amount of potassium (2.1 g/100 g), followed by calcium (0.9 g/100 g) and magnesium (0.3 g/100 g). Considering the obtained results of micromineral values in our study (iron: 567.50 ± 12.50 mg/kg, copper: 74.90 ± 0.60 mg/kg, manganese: 37.55 ± 0.45 mg/kg, zin: 16.80 ± 0.70 mg/kg, and chromium: 1.53 ± 0.04 mg/kg) and FDA recommendations [53], it can be concluded that when considering the recommended daily intake of microminerals, CS in certain quantities can successfully meet the daily needs of the human organism for microminerals. Moreover, scientists had examined many minerals in CS, among which were the microminerals examined in this paper. They showed that CS contains 843.3 mg/kg of iron, 63.3 mg/kg of copper, 50.0 mg/kg of manganese, 22.3 mg/kg of zinc, and 1.59 mg/kg of chromium [5]. These results follow the same trend as the results in this work, although slightly higher amounts of microminerals are detected in their sample, except for copper, which is higher in the CS sample analysed in this work. The same authors showed that CS contains a much higher amount of minerals compared to the coffee grounds that remain after coffee preparation, which is an indication that the skin as a by-product of coffee is one of the richer sources of minerals. Additionally, other studies, in comparison to ours, showed a higher content of copper and lower contents of iron, manganese, zinc, and chromium [7,17]. Another study showed similar quantity of chromium while the amounts of iron, zinc, and manganese were higher, and the amounts of magnesium and copper were lower [51]. When comparing our study results regarding valuable mineral quantity to the study by Martuscelli et al. [52], it was evident that except for copper and zinc which had a very similar value, and chromium which had four times smaller quantity in our study, other minerals (potassium, calcium, magnesium, sodium, iron, and manganese) were much higher in our CS study.

Furthermore, dietary fibre (DF) is one of the main dietary factors contributing to customers’ well-being. In this study, about 30% of crude fibres (31.97 ± 0.61 g/100 g) were found. Since it is only part of the insoluble fibre, the total fibre content is present in much higher quantity. Narita and Inouye [54] in their work found that CS contains 50–60% dietary fibre, of which 15% is soluble fibre and 85% is insoluble fibre. Insoluble fibres are dominated by cellulose (18%) and hemicellulose (13%) [55]. In addition, it was demonstrated that a high amount of total fibre (54.11 g/100 g) is dominated by insoluble fibres. In comparison to our study, higher amounts of insoluble fibre (46%) in mixed CS samples (*C. arabica* and *C. canephora*) were found [2,5]. Considering the large amounts of fibre, CS has great potential to be used as a raw material in the development of functional food. A recent study summarised the applications of CS-rich ingredients from various sources (e.g., bread, cakes, biscuits, cookies, yoghurt, and burger patties), where many of them replace a certain amount of flour with SC. Thus, this food was characterised as a food with reduced calories and a higher fibre content. Namely, it is very important because standard Western diet consists of too little fibre but too much calories [8]. Additionally, the same study presented a current overview of the composition of macronutrients and micronutrients, other plant secondary compounds, and contaminants; summarised other possible and valuable applications of CS; and concluded that the toxic effects of CS were not found or expected by evaluating published results of toxicity data [8].

In our study, MW procedure was optimised to obtain the highest concentration of total proteins. Under the conditions of 9 min and power of 200 W during microwave extraction, the highest concentration of extracted total proteins of 0.77 ± 0.07 mg/L in the CS samples was achieved. This concentration of 0.77 mg/L, i.e., 0.007%, in our study is a much lower concentration of proteins in CS. Other studies reported a concentration of proteins ranging from 14 to 19% in CS, which is more than the proportion of proteins in the husk or pulp of coffee beans [5,56]. The statistical processing of the experimental data of MW led to the conclusion that the duration of the extraction, the power of the microwave, and their interrelation are statistically significant factors in protein extraction. In addition to the importance of protein quantity in CS, other studies showed that a small amount of free phenol compounds is present in CS. However, CS has a marked antioxidative activity, which can be attributed to the great amount of melanoidins as Maillard reaction products [55,56]. Furthermore, static batch culture fermentation experiments showed that CS induces the preferential growth of bifidum bacteria rather than clostridia and *Bacteroides* spp. Thus, CS can be proposed as a new potential functional ingredient in consideration of its high content of soluble DF, its marked antioxidant activity, and its potential prebiotic activity [57].

Regarding the classification of CS by the European Union (EU) as a novel food [58] and considering that applications must meet legal requirements regarding composition and toxicity data [59,60], studies on its mutagenic potential are still lacking [58] and many studies nowadays have made scientific efforts to fulfil that goal [2,7,8,17,52,61]. We think that this work could contribute to this regard.

CS can be also used in the cosmetic industry [55] according to the chemical characterisation of coffee residues and their use as potential bioactive ingredients for the development of novel functional products, combined with modern needs in the pharmaceutical and related industries. The cosmetic industry is actively searching for new active ingredients due to the consumer demand for more natural and environmentally friendly products obtained through sustainable resources that can improve the skin’s appearance and its health. CS is a very interesting potential candidate to replace synthetic chemicals as an active ingredient in cosmetic formulations due to its high antioxidant potential, phenolic compounds, melanoidins, and caffeine contents. CS is a coffee-roasting by-product that is produced in large amounts every year. Its use can be seen as a gain not only for human health, but also for the environment and industries, which can answer to the principles of sustainability and circular economy. Skin aging is a complex process associated with oxidative metabolism and reactive oxygen species (ROS) generation. CS is a promising matrix that can provide the new fields of cosmetic industry a natural active ingredient that can improve the skin’s appearance and health, and counteract skin aging and related diseases, in an environmentally eco-friendly approach. One concern is the presence of ochratoxin (produced by *Aspergillus ochraceus* and *Penicillium verrucosum*) in CS. However, good practices of coffee harvesting, storage, and transport (especially related to moisture and temperature exposure) and a rigorous quality control could minimise the presence of this mycotoxin [38]. Due to CS’s richness in specific bioactive compounds (chlorogenic acids: 1–6%, caffeine: 0.8–1.25%, and melanoidins: 17–23%, among other antioxidants) and confirmed bioactivity in the prevention and/or attenuation of skin aging and related diseases (anti-inflammatory, antimicrobial, anti-cellulite, and anti-hair loss activities, and UV damage protection), CS and its extracts emerge as potential new ingredients for the cosmetic formulation sector [55]. In the modern world, overweight and obesity are the major cause of the metabolic syndrome, which is increasing rapidly in modern societies. As lifestyle changes are insufficient to improve the quality of health, nutraceuticals from the aspect of functional food from natural sources have attracted much attention as potential therapeutic agents in the prevention and treatment of obesity and other chronic diseases, including cardiovascular diseases and diabetes [62].

## 5. Conclusions

Based on the obtained results, all investigated target compounds (pesticide residues and heavy metals) are below the maximum permitted levels in food and do not pose a risk to human health. As shown in our results, due to its richness in fibre, its high amount of total proteins extracted by an optimised, eco-friendly method, MW, and its high content of valuable macro- and microminerals, CS could be used in the production of functional foods, food supplements, and cosmetics. We believe that using CS as a novel food even more widely than officially determined can provide joint optimistic approaches to extend investigation further and invest in additional efforts for stronger and more capable production based on this by-product. These approaches would reduce the amount of waste, increase the sustainable development of production, and reduce the amount of mass exploitation of biological resources and the negative impact on the environment.

## Figures and Tables

**Figure 1 foods-12-00518-f001:**
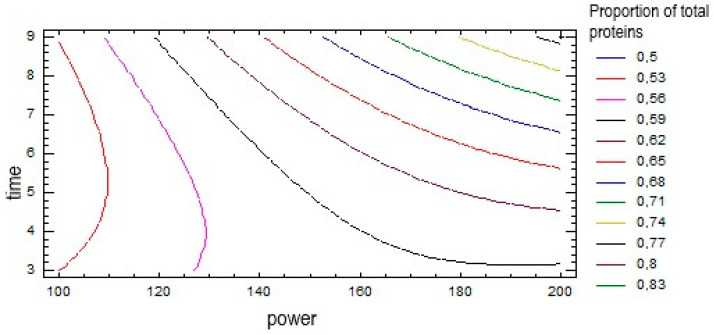
Contour plot of response surfaces for total protein concentration.

**Table 1 foods-12-00518-t001:** Chromatographic conditions of GC-MS/MS and LC-MS/MS.

	GC-MS/MS	LC-MS/MS
Instrument	GCMS-TQ8050 NX, Nexis GC-2030, Shimadzu	UPLC-MS/MS, Xevo TQ MS, Waters
Column	SH-Rxi-5Sil MS 30 m, 0.25 mmID; 0.25 um df, Restek	ACQUITY UPLC BEH 150 × 2.1 mm; 1.7 µm, Waters
Mobile phase	He	A: 5 mM ammonium formate in a mixture of water and methanol (9:1) B: 5 mM ammonium formate in methanol
Flow	1.4 mL/min	0.4 mL/min
Temperature of columns	105–290 °C	50 °C
Temperature of injectors	290 °C	10 °C
Temperature of ion source	250 °C	150 °C
Injection volume	1.0 µL	30 µL
Total run time	40 min	21 min

**Table 2 foods-12-00518-t002:** Chromatographic conditions of ICP-MS.

Atomiser	MicroMist
Injector	quartz
Cones	Nical
Rf-power	1180 W
Plasma gas flow	15.0 L/min
Atomiser gas flow	1.07 L/min
Auxiliary gas flow	0.90 L/min
Integration time	1000 ms
Points per spade	100
Number of replicas	5

**Table 3 foods-12-00518-t003:** Concentration of detected pesticides in a CS sample.

No.	Pesticide	Concentration (mg/kg)
1	Flutriafol	0.012 ± 0.006
2	Imidacloprid	0.016 ± 0.008
3	Piperonyl butoxide	0.002 ± 0.001

**Table 4 foods-12-00518-t004:** Concentration of detected heavy metals in a CS sample.

No.	Heavy Metal	Concentration (mg/kg)
1	Nickel (Ni)	2.495 ± 0.055
2	Lead (Pb)	0.249 ± 0.077
3	Arsenic (As)	0.107 ± 0.018
4	Cadmium (Cd)	0.106 ± 0.002

**Table 5 foods-12-00518-t005:** Total protein concentration (mg/L ± sd) in a CS sample treated with microwave power.

Samples	Treatment Time [min]	Power [W]	Total Proteins [mg/L ± sd]
M1	3	100	0.53 ± 0.00
M2	6	100	0.52 ± 0.01
M3	9	100	0.52 ± 0.08
M4	3	150	0.57 ± 0.00
M5	6	150	0.59 ± 0.16
M6	9	150	0.69 ± 0.01
M7	3	200	0.59 ± 0.01
M8	6	200	0.66 ± 0.06
M9	9	200	0.77 ± 0.07

**Table 6 foods-12-00518-t006:** Results of the test of statistical significance (ANOVA) of each factor of microwave extraction (MW) on the change in the concentration of total proteins.

Source	Sum of Squares	Degrees of Freedom	Mean Square	F-Ratio	*p*-Value
A: Power	0.0345649	1	0.0345649	158.04	0.0011
B: Time	0.0139394	1	0.0139394	63.74	0.0041
AA	0.000701876	1	0.000701876	3.21	0.1711
AB	0.00897756	1	0.00897756	41.05	0.0077
BB	0.000862509	1	0.000862509	3.94	0.1412
Total error	0.000656115	3	0.000218705	/	/
Total error (updated)	0.0597024	8	/	/	/

R-square = 98.901%; R-square (updated for degrees of freedom) = 97.0694%; Standard deviation, estimated = 0.0147887; Standard absolute deviation = 0.00693025; Durbin–Watson statistic = 1.63376 (*p* = 0.1962); Lag 1 residual autocorrelation = 0.162683.

## Data Availability

Data is contained within the article.

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
