# Peer review of "Coffee Silver Skin—Health Safety, Nutritional Value, and Microwave Extraction of Proteins"

_foods, 2023, doi:10.3390/foods12030518_

Round 1

Reviewer 1 Report

Dear authors

The manuscript regarding the health safety analysis of the coffee silver skin (CS) is definitely interesting, touching a topic which is very current and of particular interest nowadays, the recover of food by-products. 

There are anyway many points to be improved. 

- The abstract must be improved, showing the main obtaned results.

- tha manuscript is very similar to that one of Martuscelli et al. , published in Foods as well, with the important dfference of the MW extraction process. Therefore, in my opinion it should be more interesting underline the extraction process and the obtained results, while in my opinion this aspect is not properly treated.

- In material and methods section, it is not clear how many samples you are using. In every section of the paragraph, the number of sample is changing, and it is confusing to understand properly the kind of used coffee sample. In table 5 you talk about  6 samples but in next section, samples are 9 (M9).... and moreover, what do you mean with briquettes?  

- in my opinion it would have been more interesting to doma difference between arabica and robusta coffee for these analysis, instead to have a sample with mixed variety, with not clear proportion. which is the real provenance of the coffee?

- presented data must be improved. In table 5 you show 6 samples , 3 of them present the same value... and there is no statistics for significance. The muast be presented in a better way. The same for the table 6. Too many samples with a very similar value without statistics.  the Anova analysis is  reported in another table but in this way is not effective for understanding the results. 

I consider this topic interesting and overall the paper is well written, but I suggest to the authors to improve the paper and submit it again with corrections and a general improvement of the manuscript regardong the data presentation. 

Author Response

Dear reviewer,
The authors would like to take this opportunity to express their gratitude to the reviewers’ constructive comments and advice. All suggestions are more than welcome, especially when they amplify the quality of the work. Each reviewer comment has been discussed separately followed by the answers. All accepted remarks have been incorporated directly into the final version of the text and are marked red. Please see attachment. Kind regards.

Reviewer 2 Report

This paper deals with CS characterization and MW protein extraction of this by-product. The interest of the work is justified by discussing the potential benefits of CS according to the available literature. To my mind, the paper offers little novelty or new information on the matter of research. After carefully reading the paper, I believe this research is closer to a preliminary work, or preliminary assays. Discussion is based, to a significant extent, on others’ results, so that own results do not add significant value to the paper. Characterization is rather poor. As for microwave extraction, this could be interesting if for example compared to other extraction technologies, as for yield or quality of the protein extracted, but this is not the case.

Authors may find some more comments below.

Abstract

Reconsider/rewrite this sentence “The nutritional potential of coffee by-products, but also most 16 other by-products from the food industry, must encourage the food industry to exploit and reuse 17 these by-products”

products-nutraceutical?

The aim of this article is -> “the aim of this research was to…”

There are no results obtained in the abstract.

Introduction

Line 50. Reconsider not giving the exact value of chorogenic acid, since this is one experimental result of a particular reference. It can maybe be said, around ---, or simply state that the product is reach in this phenolic constituent.

Line 51 Which are …

Materials and methods

Line 197 as described in the introduction/as described previously

Figure 11. Calibration lines are not that relevant to be shown as figures. Remove.

Results

CS characterization is rather poor. Only one type of sample is being analyzed (CS) with no batches or differences in processing. When providing characterization results, only a few results are given (3 pesticides, 4 heavy metals, 1 value of crude fiber). It is not clear how many samples have been analyzed. There is not statistically treatment of the results (no significance of results, no homogeneous groups). See also comments below.

3.1 Pesticide residues

It is said that 265 pesticides were quantified; however, only 3 of them are detailed. Please provide reasons for that or more information regarding this decision.

Table 3 says: concentration … in a CS sample. Only one sample?? How many were analyzed?? Why is one standard deviation 0.000?

Line 235-236 What do you mean with “Imidacloprid is the next pesticide detected.”.. and “The third and last detected pesticide is piperonyl butoxide. ” Is that order relevant?

Line 244-246 Please rewrite, please provide values

3.2 Valuable minerals and heavy metals

Table 4. Again, it is said in a CS sample. Only one sample was analysed?

Line 282. Arsenic concentration is already given in table, there’s no need to repeat the value.  What about reference values in the same type of samples?? Only values in rice are discussed.

Line 319. Please remove sentence “Crude fiber was determined in studied CS simples” from the results section

Table 5. No need for showing 6 repetitions of the measurements. This table should be removed and result 31.97+-0.61 indicated in text.

This crude fiber determination is a rather simple determination. Discussion is most based on other authors’ results than in own results.

3.4. Microwave extraction

Table 6. Means are of how many repetitions? To my mind, this ANOVA results are not usually displayed like this in this kind of scientific journals. Significance of factors should be indicated in the text and discussed (multifactor ANOVA, interactions); on the other hand, homogeneous groups (from a simple ANOVA) helps identifying which mean is different from each other (in the table).

Information in paragraph 342-355 is exactly the same as in Table 6. This is not correct. If the table shows the results obtained, the text should indicate trends or main results and discussion (in this case, in a separated section). Most of the lines in this paragraph are not adding any more information than the given in the table.

What is exactly the aim of MW treatments?? Is it optimization for planning further experiments? Is it evaluating this technique as compared to other extraction methods? Is it determining the amount of protein in this by-product?

Figure 2 is of very low quality and can not be properly read.

Discussion

Line 447 (0.77+-0.07) and line 449 (14-19%), are these values comparable?? How should the reader understand this comparison?

Line 457-459. CS is proposed as a new functional ingredient, “CS can be proposed as  a new potential functional ingredient in consideration of the high content of soluble DF, the marked antioxidant activity, and the potential prebiotic activity”. But this is completely based on other authors’ research. Since none of this has been determined in this paper.

Author Response

(The authors gave the same response as above.)

Reviewer 3 Report

The following revisions should be considered:

Introduction: there are several more and more recent papers on silver skin that need to be considered, also in the discussion (e.g. see Ref. 41 for a complete review)

Line 35: I believe that coffee cherry material (up to 50% of cherry) is much larger than silver skin (1-2%).

Line 105: the botanical species is canephora not robusta

Results section: the results section already includes a discussion, and this is often reiterated in the discussion section later. I suggest to present only results in the results section, and move all discussion (i.e. comparison to literature or interpretation of heavy metal contents) to the discussion section 

Line 226: coffee sample of silver skin sample?

Section 3.4.: could you specificy the protein content in g/100g in reference to the dry silver skin as well?

Line 418: it is unclear how the daily needs were calculated? What amount of silver skin do you suggest to consume per day? 

Conclusion: the need for novel food approval in the EU could be pointed out (see Ref. 41)

Author Response

(The authors gave the same response as above.)

Round 2

Reviewer 1 Report

The authors definitely improved the manuscript addressing most the observationans and suggestion. Anyway it is still poorly faced the issue: why the choosed to use the MW extraction system without comparison with another sustainable extraction syste like, for example, ultrasound... 

Author Response

my response is in the attachment.

Reviewer 2 Report

General

Authors have answered to the specific comments I listed, but not to the introductory comment which described my impression on the paper. Therefore, I repeat this comment next for the authors to respond, if considered appropriate: “To my mind, the paper offers little novelty or new information on the matter of research. After carefully reading the paper, I believe this research is closer to a preliminary work, or preliminary assays. Discussion is based, to a significant extent, on others’ results, so that own results do not add significant value to the paper. Characterization is rather poor. As for microwave extraction, this could be interesting if for example compared to other extraction technologies, as for yield or quality of the protein extracted, but this is not the case.

I understand the paper has undergone improvements, thus authors can make reference to some of this improvements since this was an impression on the original manuscript.

Abstract

Some more information has been added in the abstract, as required by other reviewer, mainly regarding specific results. In my opinion, some of these results might be too specific. However, even if this is maintained which could be correct, the point is that some of the results which have been introduced are now after main conclusions, which is not correct.

MW extraction

Authors did not understand my comments regarding this method used. I was not asking about the general interest of MW, but indicating that the purpose of MW treatments are not properly explained in the paper. That’s the reason of the questions I raised: What is exactly the aim of MW treatments?? Is it optimization for planning further experiments? Is it evaluating this technique as compared to other extraction methods? Is it determining the amount of protein in this by-product? All these questions were raised in the context of the paper, not of the technology itself.

I have seen that other reviewer also did a similar observation when asking to “underline the extraction process and the obtained results, while in my opinion this aspect is not properly treated”. This comment is in line with my comment, and I would insist on the authors giving more relevance to the extraction process, comparing with other techniques, etc.  

Language revision

Please revised the final text. Typos and repetitions should be eliminated.

Author Response

Dear reviewer,

Thank you for your comments and suggestions.

Please see the attachment with our answers.

Kind regards.
